# Development, and Internal, and External Validation of a Scoring System to Predict 30-Day Mortality after Having a Traffic Accident Traveling by Private Car or Van: An Analysis of 164,790 Subjects and 79,664 Accidents

**DOI:** 10.3390/ijerph17249518

**Published:** 2020-12-18

**Authors:** Antonio Palazón-Bru, María José Prieto-Castelló, David Manuel Folgado-de la Rosa, Ana Macanás-Martínez, Emma Mares-García, María de los Ángeles Carbonell-Torregrosa, Vicente Francisco Gil-Guillén, Antonio Cardona-Llorens, Dolores Marhuenda-Amorós

**Affiliations:** 1Department of Clinical Medicine, Miguel Hernández University, San Juan de Alicante, 03550 Alicante, Spain; folgado.david@gmail.com (D.M.F.-d.l.R.); ana.macanas@goumh.umh.es (A.M.-M.); emmamares@hotmail.com (E.M.-G.); carbonell_mar@gva.es (M.d.l.Á.C.-T.); vgil@umh.es (V.F.G.-G.); 2Department of Pathology and Surgery, Miguel Hernández University, San Juan de Alicante, 03550 Alicante, Spain; antonio.cardona@umh.es (A.C.-L.); d.marhuenda@umh.es (D.M.-A.)

**Keywords:** accidents, traffic, mortality, death, models, statistical

## Abstract

Predictive factors for fatal traffic accidents have been determined, but not addressed collectively through a predictive model to help determine the probability of mortality and thereby ascertain key points for intervening and decreasing that probability. Data on all road traffic accidents with victims involving a private car or van occurring in Spain in 2015 (164,790 subjects and 79,664 accidents) were analyzed, evaluating 30-day mortality following the accident. As candidate predictors of mortality, variables associated with the accident (weekend, time, number of vehicles, road, brightness, and weather) associated with the vehicle (type and age of vehicle, and other types of vehicles in the accident) and associated with individuals (gender, age, seat belt, and position in the vehicle) were examined. The sample was divided into two groups. In one group, a logistic regression model adapted to a points system was constructed and internally validated, and in the other group the model was externally validated. The points system obtained good discrimination and calibration in both the internal and the external validation. Consequently, a simple tool is available to determine the risk of mortality following a traffic accident, which could be validated in other countries.

## 1. Introduction

Road traffic accidents are a serious public health problem, with estimates using data from recent decades suggesting they could become the fifth leading cause of death worldwide by 2030 [1,2]. In addition, traffic accidents are an important socioeconomic problem because of the need to treat victims, the loss of productivity of those who die or are disabled as a result of the accident, the years of life lost and the time spent by families caring for victims due to the sequelae [3].

Attempts to reduce the number of accidents have involved great effort but have nevertheless met with very varied results, depending partly on the idiosyncrasy and culture of the countries involved [4,5]. Although specific measures associated with comprehensive intervention packages or more complex regulations have had a considerable impact in terms of reducing the number of accidents, they have had less impact on the reduction in the number of victims (driver/passenger) [6]. They have, however, had a very important effect on the decrease in the mortality rate of pedestrians involved in accidents. This aspect is closely associated with the specific country in which the study is carried out [7].

Factors that are decisive in relation to mortality in traffic accidents include driver-dependent factors, such as being male and young, or environmental factors, such as low brightness, dawn or dusk, and adverse weather conditions. In addition, the most frequent accidents involve pedestrians being struck or run over, or head-on collisions, the most common causes of which are negligence or recklessness, in both industrial and emerging countries [8,9,10,11,12,13,14,15,16,17]. Obesity, as a driver-dependent factor, has also been found to significantly increase the risk of death in the event of a traffic accident [18].

Analysis of transport data [19,20,21,22,23,24,25] has led to detailed study of the association of accident density and severity, assessing various methodologies and such factors as congestion, flow, speed, and road geometry (curvature and gradient), which are all relevant for the consequences of an accident. The results of these studies show that a greater density of vehicles (vehicles per hour) is not directly associated with greater severity, after controlling for the previously mentioned confounding factors. In addition, with these methodologies we can determine road segments where the probability of having a fatal traffic accident is higher.

However, just knowing the factors associated with mortality does not provide information about the specific weight of each factor in the prediction. This is where prediction models play a key role, since they determine, through complex mathematical formulas, the probability that a subject with certain characteristics will experience a certain event, in our case that the subject will die as a result of a traffic accident.

In 2014, an international consensus of experts on predictive models was published, indicating which aspects should be taken into account when developing a new model or validating an existing model when carrying out a systematic review of predictive models [26]. Subsequently, in 2019, a quality assessment system was established for predictive models, similar to that of the Cochrane tool for randomized clinical trials, to assess the risk of bias and applicability (Prediction model Risk Of Bias ASsessment Tool, PROBAST) [27,28]. These statements are based on the best available scientific evidence, and following them is essential to obtain a prediction model that can be used in practice and thereby take appropriate measures to reduce the risk of a subject experiencing the event of interest [27,28].

In the scientific literature, we found only one study that developed a model for predicting traffic accident mortality [29]. In this publication, there were only 54 events (people killed in the accident) out of a total of 3922 people involved in accidents who had gone to the emergency department of a hospital in Belgium in 2008–2011. The individuals who sustained no injuries during the accident were not included in the study. Furthermore, adherence to the recommended statistical methods was not high, which increases the risk of bias and reduces applicability [27,28].

In order to develop and both internally and externally validate a prediction model to determine 30-day mortality after a traffic accident, all private car and van accidents occurring in Spain in 2015 were analyzed, involving 164,790 subjects and 79,664 accidents. All analyses were performed applying the methodology indicated since 2014 [26,27,28]. This provided a tool whereby we can see the specific weight of each factor to produce a traffic accident fatality, and thus determine where we need to intervene. In addition, after the publication of the prediction model, we intend to integrate it into a mobile application for Android, which will be free for all Google Play users (“Traffic accident mortality”). This app could be used by the police and emergency services, as well as hospital emergency teams, to estimate the mortality risk directly and improve decision making.

## 2. Materials and Methods

### 2.1. Setting

Since 1993, the Directorate General of Traffic, with the collaboration of the Directorate General of the Civil Guard and the autonomous and municipal police, all have the obligation to collect, at least annually, records of all traffic accidents occurring in Spain. Each of these collections must include a series of explicit characteristics, such as the type of road, the number of vehicles involved, the place, the type of accident, the most important determining factor of the accident, and whether anyone died [30]. The individual accident questionnaires must be completed by the agents in charge of traffic surveillance and control, who, after the appropriate review and in order to avoid possible omissions or errors, send them directly to the corresponding Provincial Traffic Headquarters within five days of the accident. Since that same year, these data are computerized and are available to the public, along with all the reports prepared by the Directorate General of Traffic. However, the latest data as of June 2019 are from the year 2015 and these, therefore, have been used to carry out this study.

The Directorate General of Traffic defines an accident with victims as one in which the following circumstances are present: it occurs on or has its origin on a road or land subject to the legislation on traffic, motor vehicle circulation, and road safety; it also involves at least one vehicle in motion, and as a result, one or more persons are killed and/or injured. Thus, any person who dies immediately or within thirty days of a traffic accident is deemed to have died as a result of a traffic accident. Confirmed cases of natural death or suicide are excluded. These definitions are contained in Order INT/2223/2014, of 27 October, which regulates the communication of information to the National Register of Traffic Accident Victims [31].

### 2.2. Study Design and Participants

The study design involves a cohort of subjects traveling in a private car or van who were followed from the time of having an accident with victims in 2015 in Spain up to a maximum period of 30 days. Mortality is reflected in the database in binary form, unless the data is unknown, in which case a probability of death is indicated. The 2015 data were complete, which implies that there was no loss to follow-up.

### 2.3. Ethical Aspects

All data are free and open access, and completely anonymized, identifying each person as well as the accident with a unique code. The data analysis was approved by the Project Evaluation Body of Miguel Hernández University with reference DPC.ALC.01.19 on 11 April 2019.

### 2.4. Variables and Measurements

The main study variable was death within 30 days of the accident. This is the period recorded by the Directorate General of Traffic, though no indication is given of the exact date of death within this period after the accident. Note that this variable is completely objective and knowledge of it would not interfere with the measurement of the predictors (blinding of outcome), which are detailed below. The selected candidate predictors were those factors concerning the accident that could affect mortality with a completely objective and simple measure: (1) Accident-related variables: weekend (yes, no; starting on Friday at 12 noon), time of the accident (hour), number of vehicles in the accident, road (urban, interurban), brightness (daylight, dusk, night (sufficient lighting), night (insufficient lighting)) and weather (good, fog, rain or hail, snowing, high wind); (2) Vehicle-related variables: type (private car, van) and age of the vehicle (years), and other vehicles in the accident (private car, van, truck, bicycle, motorbike or moped, heavy equipment or tractor, bus); and (3) People-related variables: gender (male, female), age (years) and using a seat belt (yes, no). All variables were obtained from the database described above. Since all predictors are objective, bias due to lack of blinding of predictors is completely minimized. In addition, all predictors are known at the time of an accident involving victims.

### 2.5. Sample Size

To develop a prediction model, according to the PROBAST initiative, at least 20 events-per-variable are necessary, including variable transformations, while to externally validate a model, 100 events, and 100 non-events are required [27,28]. Our sample was divided into two groups, one to develop and internally validate the model and the other to externally validate it. This division was done using the accidents that occurred in odd months for development and internal validation, and the rest for external validation. This process was not carried out in a completely random manner, as it is recommended to divide the samples by temporal criteria, location, etc. [26]. The first sample consisted of 84,502 people and 399 deaths, which enabled construction of a predictive model with up to 19 predictors (399÷20), while the second sample comprised 80,288 individuals and 379 deaths, which met the standard to externally validate a predictive model [27,28].

### 2.6. Statistical Methods

Categorical variables were described using absolute and relative frequencies, while quantitative variables were described using means and standard deviations. The descriptive analysis was performed stratifying by group (development/internal validation and external validation). The missing data in some of the predictors were imputed, as recommended in PROBAST [27,28], using multivariate imputation by chained equations [32]. The continuous predictors were transformed by means of cubic splines, to study the functional form of the variable [33]. The knots for age were 25, 50, and 75 years, for time of occurrence of the accident 06:00, 12:00, and 18:00, and for age of the vehicle 5, 10, and 15 years. The knot points were pre-specified in order to divide the complete interval into similar parts. Firstly, we compared the spline models with ones assuming a linear form for the relationship, determining whether or not spline modelling is necessary. This was done performing graphs of the relationship between time, person age, vehicle age and outcome. We selected the spline functions when graphical differences were apparent between the two curves. The process of converting a variable via splines equates to that variable being transformed into 6 independent spline variables (this will depend on the number of knots selected). Each of these variables is then considered as a candidate predictor in the construction of the model, and there may be some that are significant and some that are not. After this, all the variables were introduced in a binary logistic regression model, substituting the continuous linear variables for spline transformations, which totaled 41 predictors. It was not possible, therefore, to introduce all the predictors in the model and the final list was selected using a forward method based on the likelihood ratio test (*p*-value < 0.25 as the criterion to be included). The mathematical model, which is difficult to use in practice, was adapted to a points system (Framingham Heart Study methodology), which weighs the coefficients and groups the values of the predictors to make it much easier to apply, although accuracy is lost [34]. Following this transformation, the model was internally validated by bootstrapping, determining both discrimination and calibration. This procedure consists of taking many random samples with replacement of the original sample (generally 1000) and through them constructing the distribution of a statistic, thus calculating its mean, standard deviation, etc [35]. Discrimination was evaluated by calculating the area under the receiver operating characteristic curve and calibration by means of a calibration plot, obtaining the observed probabilities of death by means of cubic splines, as recommended in the literature [36]. The same method was applied for the external validation, using the sample comprising accidents that occurred in even months, evaluating the area under the receiver operating characteristic curve (AUC) and the calibration plot. All analyses were performed with a significance of 5% and for each relevant parameter its associated confidence interval (CI) was obtained. The statistical packages used were IBMS SPSS Statistics 25 (IBM Corporation, Armonk, NY, USA) and R 3.5.3 (R Foundation for Statistical Computing, Vienna, Austria).

## 3. Results

The development and internal validation sample comprised a total of 41,082 accidents and 84,502 individuals, while the external validation sample consisted of 38,682 accidents and 80,288 individuals. Great similarity was observed between the two samples in the accident-related variables (Table 1), highlighting that a quarter of the accidents occurred on weekends, a third on interurban roads, generally with good lighting and good weather. We also observed this homogeneity in the people-related variables, with a similar mortality (~0.5%), a majority being men, with a mean age close to 40 years and the majority of vehicles being private cars. The number of missing data for each sample and predictor, which were later imputed, is also shown in Table 1. Finally, the transformation of continuous predictors is presented in Appendix A. We selected the spline functions based on the results obtained in Appendix A (great differences in all cases).

The model coefficients, together with their standard errors and *p*-value, are shown in Table 2, with significantly higher mortality (*p* < 0.05) associated with fewer vehicles involved, interurban roads, poor visibility, men, and people not wearing a seat belt. Mortality was greater when a truck or a bus was involved in the accident, and less with another private car, motorbike or moped. The adaptation of this model to the points system is depicted in Figure 1.

Regarding validation, the discrimination is shown in Figure 2 and the calibration in Figure 3 and Table 3. The AUC (discrimination) had a mean value of 0.88 in the internal validation and 0.87 in the external validation (standard deviation of 0.1 in both cases). The smooth calibration curve is illustrated in Figure 3 and may appear to have a problem with fit in the external validation. However, we have indicated all the probabilities (observed and expected) and frequencies (cumulative and absolute) in Table 3, where we see that the difference between the observed and expected probability is almost always less than 5% (99.9% of the cases) and a greater deviation is seen only in the cases with high scores, though this is highly unlikely (this would be an accident with all the conditions with the highest scores: no seat belt, fewer than four vehicles, interurban road, etc.) as they would be subjects who would sum nearly the maximum possible score in all the predictors. In other words, the model has a good fit in all the scores for the internal validation and for the external validation when the score value is between −5 and 3, with a greater margin of error when the subject has 4 (6.81%) or 5 (18.91%) points. With these scores the model appears to adjust incorrectly, especially with 5 points, but we are talking about 49 subjects out of a total of 80,288.

## 4. Discussion

### 4.1. Summary

This study was developed and validated, both internally and externally, on a national sample of about 165,000 individuals, with a model that predicts 30-day mortality for the victim of a traffic accident traveling by private car or van. This model has been adapted to a points system to increase its applicability and will be integrated into an Android application, which will allow the immediate calculation of the risk.

### 4.2. Strengths and Limitations

The main strength of this study is the innovative idea developed, which is weighing the already known mortality factors in a single model [8,9,10,11,12,13,14,15,16,17] that is very easy to use. In addition, by following all the PROBAST requirements to construct our model [27,28], the risk of bias was reduced, and the applicability increased. Finally, a sample size of approximately 165,000 people gave our results high precision.

Selection bias was minimized since, by law, all accidents involving victims must be recorded by the appropriate bodies, which provides maximum representativeness of the population, to which is added the large number of individuals. Concerning information bias, we must accept this in our study to a certain extent as we worked with data collected retrospectively. However, the percentage of missing data was minimal and the variables are completely objective, with no possibility of interpretation by the police. Moreover, confusion bias was minimized by applying a multivariate model with the most recommended techniques and assessing the functional form of the variables age and time of the accident. Another limitation could have been the lack of inclusion of certain factors related to the vehicle, like the airbag, or more general factors (driver response, alcohol/drugs, types of road with more subtypes than just interurban or urban roads, types of road section (bridge, tunnel, embankment, trench, etc.) or collision and speed limit), as these are not registered in the database. However, the good discriminative capacity and calibration mean that we can obtain predictions even without these other factors. Nonetheless, if more information was available about the more general variables we could improve the precision of the model, which is in fact already very satisfactory. Finally, the model is strongly conditioned by the specific country analyzed in our study, and for that reason should be externally validated in other countries. However, taking into account these limitations, we could design other studies with primary sources adding more relevant prognostic factors, in order to achieve better accuracy to make predictions and, consequently, take measures to prevent a fatal accident.

### 4.3. Comparison with the Existing Literature

Comparing our results poses difficulties because, to the best of our knowledge, no predictive models with these characteristics have been published, since the above-mentioned model [29] did not work with this type of data, and the individuals were those who attended their own hospital. Furthermore, the model of these authors did not follow the PROBAST indications, which limits its use, since as early as 2014 indications were available on the procedure to develop a predictive model [26].

Analyzing the factors found in our model, we obtained expected results: not using the seat belt, unfavorable lighting, interurban roads where greater speed is reached and small vehicles drive alongside buses or trucks, which in the event of a small vehicle rollover could cause the death of its occupants [8,9,10,11,12,13,14,15,16,17]. However, considering victim age, since its functional form was not studied by other researchers as it was in our study [8,9,10,11,12,13,14,15,16,17], we must analyze the association found more closely. This makes sense, since higher mortality appears in subjects of very advanced age, who are at greater risk, as seen in a meta-analysis published in 2018 [37]. It is also noteworthy that individuals younger than 60 years had lower mortality rates. This fact should be confirmed with other studies assessing the functional form of age in mortality.

### 4.4. Implications to Public Health and Research

Over the years, campaigns have been put in place to prevent fatalities in road accidents, such as the use of seat belts, the use of helmets for two-wheeled vehicles and blood alcohol and narcotics checks [37,38,39]. As a result of our findings, we see that there are other types of factors on which we can intervene and thus reduce traffic accident mortality, highlighting that areas with inadequate lighting or poor lighting should be corrected to avoid this problem.

Our predictive model has been validated both internally and externally. We therefore have a tool to indicate the risk of death of a person involved in a traffic accident in a private car or van involving at least one victim. This model is also very useful for clinicians who receive injured victims at the hospital, as they will be able to stratify patient risk and thus improve therapeutic performance. Moreover, the emergency coordinating center can assess the circumstances of the accident and prioritize emergency health resources for individuals at greatest risk of death.

Concerning future research on traffic accidents, a highly relevant public health topic, we recommend the external validation of our predictive model in other countries using PROBAST methodology [27,28]. All this with the aim of reducing mortality due to traffic accidents and annulling the forecast of an increase in the following decade [1,2]. On the other hand, the next studies, in addition to following the PROBAST checklist, should also consider new fundamental parameters such as types of road, types of road section, speed limits and collision speed. This will improve our model over the years and reduce the mortality risk in a traffic accident with victims.

## 5. Conclusions

A predictive model of 30-day mortality after a traffic accident with victims traveling in a private car or van has been developed and applied in a simple manner through a points system. The system considers as parameters age, use of a seat belt, and the type of vehicle in which the individual was not travelling, as well as the characteristics of the accident itself (brightness, number of vehicles involved and type of road). Overall, good validation indicators were obtained both internally and externally, and the model should be implemented systematically and be used to design interventions to reduce mortality from traffic accidents. This model has a great practical implication, as it clearly indicates the factors with the greatest weight when there is a death in a traffic accident with victims. This information could be used for both the police and the health services to make predictions, as well as to determine where to undertake possible interventions to reduce the risk of death. This would be done by analyzing the factors in the model, attempting to prevent them reaching the categories with higher scores. For example, avoiding areas with poor illumination, increasing controls to check the use of seat belts, or establishing regulations for large vehicles or on roads where faster speeds are reached. We also encourage other authors to validate our model following PROBAST methodology to determine whether it is applicable to countries other than Spain.

## Figures and Tables

**Figure 1 ijerph-17-09518-f001:**
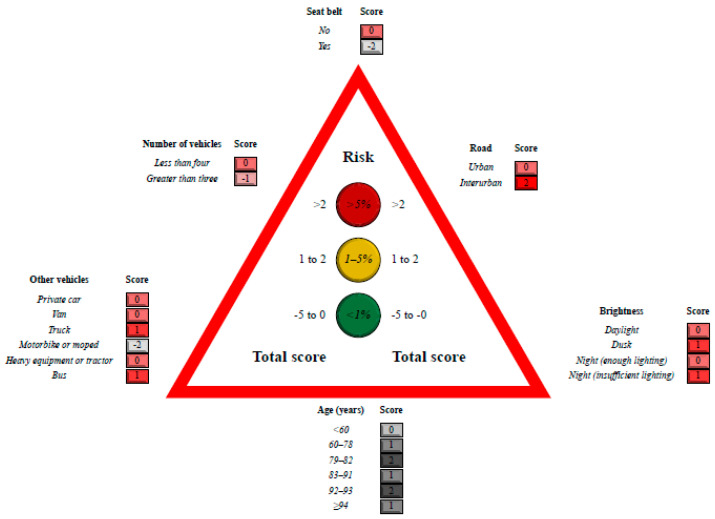
Scoring system to predict 30-day mortality in traffic accidents. Sum the score associated with each variable (around the traffic signal in red and gray colors) so as to calculate the total score and determine the risk.

**Figure 2 ijerph-17-09518-f002:**
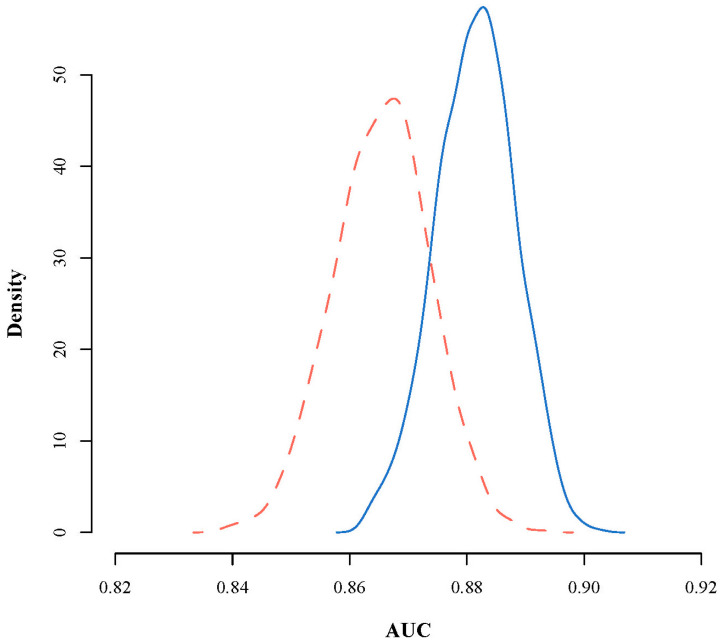
Discrimination of the scoring system using bootstrapping in the internal and external validation. AUC, area under the Receiver Operating Characteristic curve. Blue, internal validation; Red, external validation (dashed line).

**Figure 3 ijerph-17-09518-f003:**
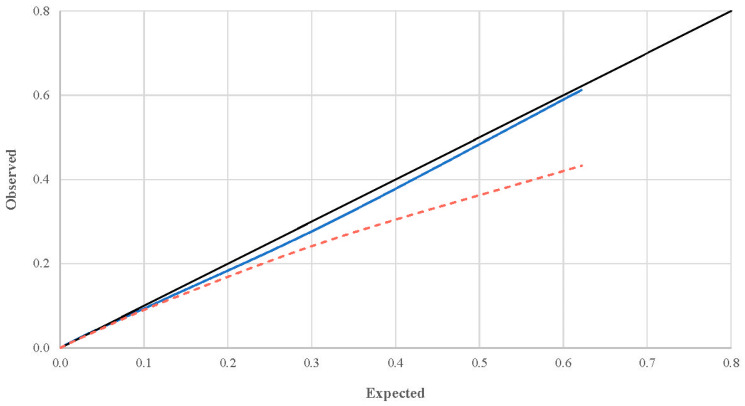
Smooth calibration (splines) of the scoring system using bootstrapping in the internal and external validation. The 30-day probability of death is represented in the graph. Blue, internal validation; Red, external validation (dashed line).

**Table 1 ijerph-17-09518-t001:** Main characteristics of the two cohorts for developing and externally validating the predictive model for 30-day mortality.

Variable	Development Cohort n(%)/x ± s	Validation Cohort n(%)/x ± s
Accidents (Development n = 41,082; Validation n = 38,682)		
Weekend:		
Yes	9567 (23.3)	8965 (23.2)
No	31,515 (76.7)	29,717 (76.8)
Time (hours) ^1^	14.0 ± 5.2	14.0 ± 5.2
Number of vehicles ^1^	1.9 ± 0.7	1.9 ± 0.7
Road:		
Urban	26,649 (64.9)	24,659 (63.7)
Interurban	14,433 (35.1)	14,023 (36.3)
Brightness:		
Daylight	29,531 (71.9)	27,605 (71.4)
Dusk	2267 (5.5)	2195 (5.7)
Night (sufficient lighting)	6728 (16.4)	6334 (16.4)
Night (insufficient lighting)	2556 (6.2)	2548 (6.6)
Weather:		
Good	34,219 (83.3)	31,818 (82.3)
Fog	280 (0.7)	292 (0.8)
Rain or hail	3036 (7.4)	2816 (7.3)
Snowing	50 (0.1)	54 (0.1)
High wind	185 (0.5)	130 (0.3)
Other	1372 (3.3)	1780 (4.6)
Missing	1940 (4.7)	1792 (4.6)
Vehicles (Development n = 59,807; Validation n = 56,302)		
Type:		
Private car	54,502 (91.1)	51,202 (90.9)
Van	5305 (8.9)	5100 (9.1)
Age of the vehicle (years):		
x ± s	10.3 ± 5.8	10.3 ± 5.9
Missing	16,580 (27.7)	15,488 (27.5)
Other vehicles in the accident:		
Private car	31,573 (52.8)	29,554 (52.5)
Van	3821 (6.4)	3732 (6.6)
Truck	2243 (3.8)	2179 (3.9)
Bicycle	2042 (3.4)	1971 (3.5)
Motorbike or moped	10,899 (18.2)	10,074 (17.9)
Heavy equipment or tractor	129 (0.2)	114 (0.2)
Bus	561 (0.9)	518 (0.9)
People (Development n = 84,502; Validation n = 80,288)		
30-day mortality:		
Yes	399 (0.5)	379 (0.5)
No	84,103 (99.5)	79,909 (99.5)
Gender:		
Male	50,488 (59.7)	47,898 (59.7)
Female	32,717 (38.7)	31,199 (38.9)
Missing	1297 (1.5)	1191 (1.5)
Age (years):		
x ± s	39.6 ± 17.1	39.4 ± 17.1
Missing	3294 (3.9)	3130 (3.9)
Seat belt:		
Yes	61,687 (73.2)	58,811 (73.3)
No	1984 (2.3)	2001 (2.5)
Missing	20,651 (24.4)	19,476 (24.3)
Position in the vehicle:		
Driver	59,731 (70.7)	56,192 (70.0)
Front passenger	14,274 (16.9)	13,708 (17.1)
Rest of passengers	10,497 (12.4)	10,388 (12.9)

Abbreviations: n(%), absolute frequency (relative frequency); x ± s, mean ± standard deviation. ^1^ no missing data in these variables. The development cohort consists of the accidents that occurred in January, March, May, July, September, and November. The validation cohort consists of the accidents that occurred in February, April, June, August, October, and December.

**Table 2 ijerph-17-09518-t002:** Predictive model for 30-day mortality after having a traffic accident.

Variable	B (SE)	*p*-Value
Number of vehicles	−0.27 (0.11)	0.012
Interurban road	2.53 (0.20)	<0.001
Brightness:		
Dusk	0.96 (0.19)	<0.001
Night (insufficient lighting)	1.03 (0.13)	<0.001
Weather:		
Rain or hail	−0.35 (0.20)	0.075
Gender male	0.43 (0.12)	<0.001
Other vehicles in the accident:		
Private car	−0.43 (0.18)	0.013
Motorbike or moped	−2.67 (0.72)	<0.001
Van	−0.48 (0.27)	0.08
Truck	1.32 (0.18)	<0.001
Bus	1.00 (0.45)	0.028
Age (years):		
S2	1.29 (0.56)	0.022
S3	1.16 (0.34)	0.001
S4	2.77 (0.54)	<0.001
S5	3.92 (0.54)	<0.001
S6	2.92 (1.43)	0.042
Time (hours):		
S5	−0.95 (0.28)	0.001
Seat belt	−2.54 (0.13)	<0.001
Age of the vehicle (years):		
S3	0.25 (0.18)	0.168
Intercept	−6.26 (0.43)	<0.001

Abbreviations: B, regression coefficient; SE, standard error; S, splines. We transformed the continuous predictors into cubic B-splines. The knots were 25, 50, and 75 years for the age variable; 6, 12, and 18 h for the time variable; and 5, 10, and 15 for the age of vehicle variable. Number of vehicles was not transformed using splines, because most of accidents involved fewer than four vehicles (97.9%). Events-per-variable rate of the model: 399/19 = 21 > 20.

**Table 3 ijerph-17-09518-t003:** Probabilities for 30-day mortality in the scoring system and calibration results using splines.

Score	Expected Risk (%)	Internal Validation					External Validation				
		Absolute Frequency	Relative Frequency (%)	Cumulative Relative Frequency (%)	Observed Risk (%)	Risk Difference (%) ^1^	Absolute Frequency	Relative Frequency (%)	Cumulative Relative Frequency (%)	Observed Risk (%)	Risk Difference (%) ^1^
−5	0.00	162	0.2	0.2	0.00	0.00	127	0.2	0.2	0.00	0.00
−4	0.00	7895	9.3	9.5	0.00	0.00	7335	9.1	9.3	0.00	0.00
−3	0.01	2809	3.3	12.9	0.00	0.01	2662	3.3	12.6	0.01	0.00
−2	0.03	30,050	35.6	48.4	0.02	0.01	27,660	34.5	47.1	0.04	0.00
−1	0.12	9496	11.2	59.7	0.10	0.01	8755	10.9	58.0	0.12	0.01
0	0.39	20,779	24.6	84.2	0.40	0.01	20,462	25.5	83.5	0.40	0.01
1	1.28	10,869	12.9	97.1	1.35	0.07	10,815	13.5	96.9	1.25	0.03
2	4.18	1925	2.3	99.4	4.15	0.03	1977	2.5	99.4	3.89	0.28
3	12.75	471	0.6	99.9	11.85	0.90	446	0.6	99.9	11.25	1.50
4	32.90	39	0.0	100.0	30.53	2.37	46	0.1	100.0	26.10	6.81
5	62.19	7	0.0	100.0	61.26	0.93	3	0.0	100.0	43.28	18.91
6	84.66	0	N/C	N/C	N/C	N/C	0	N/C	N/C	N/C	N/C
7	94.87	0	N/C	N/C	N/C	N/C	0	N/C	N/C	N/C	N/C
8	98.42	0	N/C	N/C	N/C	N/C	0	N/C	N/C	N/C	N/C
9	99.52	0	N/C	N/C	N/C	N/C	0	N/C	N/C	N/C	N/C

N/C, not calculated because we had no subjects with this score. ^1^ This is the difference in absolute value between the expected and observed risk.

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
