# Peer review of "Development, and Internal, and External Validation of a Scoring System to Predict 30-Day Mortality after Having a Traffic Accident Traveling by Private Car or Van: An Analysis of 164,790 Subjects and 79,664 Accidents"

_ijerph, 2020, doi:10.3390/ijerph17249518_

Round 1

Reviewer 1 Report

In this paper the authors analyse a population database of traffic accidents in order to develop a logistic regression model to predict the probability of mortality following a traffic accident. The strengths and limitations of the study are well described along with implications for policy

The statistical methods section is missing quite a bit of detail. Please refer to Categorical rather than Qualitative variables in the opening sentence (line 141). The multivariate imputation by chained equations (line 145) needs a reference.

The cubic splines (line 146) also need a reference. The authors should explain whether the knot points are prespecified or whether they were estimated for the data, and the reason for the method chosen. For a predictive model that is supposed to generate a simple tool for decision-makers to use, the choice of cubic splines seems unnecessarily complicated and it introduces so many more variable sin the model that other potentially useful predictors have to be left out. The authors should compare the spline models with ones assuming a linear form for the relationship, and satisfy the readers that the spline modelling is necessary. Some graphs of the relationship between time, person age, vehicle age and the outcome would also help to justify the decisions made. What does it mean for only some of the splines to be included in the model? The authors should explain what it means for, say, the 5th spline of vehicle age to be a significant predictor in the model.

The authors should explain more about the bootstrapping procedure (line 154), possibly including a reference to other researchers who have used it or to the development of the methods for this situation.

Figure 1: the authors should explain how they got from the model in Table 2 to the scoring system

Figure 3: the authors should consider whether it would be helpful to show a scatter of points to provide information about spread as well as location. The explanation of why the red line stops (line 198) is not convincing and needs to be elaborated.

In the light of these concerns, particularly around the cubic splines, I recommend that the paper undergo major revision before resubmission.

Minor comments:

Line 58: I think this paper is best described as a consensus of experts on systematic reviews of predictive models.

Line 196: why does AUC have a mean value? What are the multiple calculations of it or what does the curve look like?

Line 202: what is being referred to here that is highly unlikely?

Author Response

In this paper the authors analyse a population database of traffic accidents in order to develop a logistic regression model to predict the probability of mortality following a traffic accident. The strengths and limitations of the study are well described along with implications for policy

Thank you very much for your feedback!

The statistical methods section is missing quite a bit of detail. Please refer to Categorical rather than Qualitative variables in the opening sentence (line 141).

Information has been added and the change made.

The multivariate imputation by chained equations (line 145) needs a reference.

We have included a reference about that method.

The cubic splines (line 146) also need a reference.

We have also included a reference for this.

The authors should explain whether the knot points are prespecified or whether they were estimated for the data, and the reason for the method chosen.

We have included an explanation in the relevant place.

For a predictive model that is supposed to generate a simple tool for decision-makers to use, the choice of cubic splines seems unnecessarily complicated and it introduces so many more variable sin the model that other potentially useful predictors have to be left out. The authors should compare the spline models with ones assuming a linear form for the relationship, and satisfy the readers that the spline modelling is necessary. Some graphs of the relationship between time, person age, vehicle age and the outcome would also help to justify the decisions made.

We have followed your indications, placing three new figures as supplemental material. The changes are in Statistical methods and Results.

What does it mean for only some of the splines to be included in the model? The authors should explain what it means for, say, the 5th spline of vehicle age to be a significant predictor in the model.

An explanation has been included in Statistical methods.

The authors should explain more about the bootstrapping procedure (line 154), possibly including a reference to other researchers who have used it or to the development of the methods for this situation.

We have included more information about this procedure, including a reference.

Figure 1: the authors should explain how they got from the model in Table 2 to the scoring system

We applied the Framingham Heart Study algorithm, as indicated in the previous version of the text: “The mathematical model was adapted to a points system, which weighs the coefficients and groups the values of the predictors to make it much easier to apply, although accuracy is lost [27].” If the reviewer considers more particulars about the method are required, we can include them, but the paper is already quite extensive concerning the number of words.

Figure 3: the authors should consider whether it would be helpful to show a scatter of points to provide information about spread as well as location.

We have not done a scatter plot as there are only 11 points, which are the different values the scoring system can adopt (-5 to 5 in our data set) and it would not provide any useful additional data. This is why we have joined the points with a line.

The explanation of why the red line stops (line 198) is not convincing and needs to be elaborated.

We have included more information in order to clarify this issue.

In the light of these concerns, particularly around the cubic splines, I recommend that the paper undergo major revision before resubmission.

We have followed your suggestion so as to improve our work.

Minor comments:

Line 58: I think this paper is best described as a consensus of experts on systematic reviews of predictive models.

We have corrected this part of the text.

Line 196: why does AUC have a mean value? What are the multiple calculations of it or what does the curve look like?

As we have now explained the bootstrapping procedure, we think that this question is solved.

Line 202: what is being referred to here that is highly unlikely?

This is now indicated in parenthesis.

Reviewer 2 Report

This is an interesting paper discussing predictive factors for fatal traffic accidents, several comments on the paper:

How to determine the scale in Scoring system of Figure 1 and why?

No potential research or remediation plan in section Strengths and limitations  about how to address the limitations

The conclusion is relatively weak, it only repeated what they did, but what is the contribution of this research? 

Author Response

This is an interesting paper discussing predictive factors for fatal traffic accidents, several comments on the paper:

Thank you very much for your positive comments.

How to determine the scale in Scoring system of Figure 1 and why?

The logistic regression model, which is difficult to use in practice, has been adapted to a scoring system, using the Framingham Heart Study methodology, as we indicated in the previous version of the text: “The mathematical model was adapted to a points system, which weighs the coefficients and groups the values of the predictors to make it much easier to apply, although accuracy is lost [27].”.

In order to explain the method of determining the risk, we indicated this in the legend to Figure 1: “Sum the score associated with each variable (around the traffic signal in red and gray colors) so as to calculate the total score and determine the risk”.

The importance of applying the method has been highlighted in this new version in Statistical methods (“why?”).

No potential research or remediation plan in section Strengths and limitations about how to address the limitations

Some comments have been added in limitations, following your suggestion and those from reviewer #3.

The conclusion is relatively weak, it only repeated what they did, but what is the contribution of this research?

We have written new content in the conclusion, following your guidelines.

Reviewer 3 Report

Dear authors,

I had the opportunity to revise the paper “Development, and internal and external validation of a scoring system to predict 30-day mortality after having a traffic accident traveling by private car or van: an analysis of 164,790 subjects and 79,664 5 accidents”. In this study, the authors analysed data on all road traffic accidents with victims involving a private car or van occurring in Spain in 2015 (for a total of 164,790 subjects and 79,664 accidents) in order to obtain a simple tool which is able to determine the risk of mortality following a traffic accident.

The results achieved are very interesting and promising.

English language and style are fine.

The paper needs however some improvements to make it more suitable for publication.

The reviewer would suggest these main actions:

  • Improve the structure of the paper. In its present form the paper is rather unbalanced. Introduction occupies a too little section: although the literature review presents a good level of details, it would be appropriate to add some papers, from major international transportistic journals, on the relationship between accident density, and its severity with a variety of independent variables practically ignored in this study.
  • Highlight the limitations: the model is strongly conditioned by the specific country analysed in the study and by the absence of many other parameters that influence the gravity of a crash.
  • Strengthen the conclusions: authors should be better explaining how this implemented model can be used to design interventions to reduce mortality from traffic accidents.
  • In the next studies, it would be important to consider new fundamental parameters such as: types of road (not only urban or interurban), types of road section (bridge, tunnel, embankment, trench, etc.), speed limits and speed collision.

Author Response

Dear authors,

I had the opportunity to revise the paper “Development, and internal and external validation of a scoring system to predict 30-day mortality after having a traffic accident traveling by private car or van: an analysis of 164,790 subjects and 79,664 5 accidents”. In this study, the authors analysed data on all road traffic accidents with victims involving a private car or van occurring in Spain in 2015 (for a total of 164,790 subjects and 79,664 accidents) in order to obtain a simple tool which is able to determine the risk of mortality following a traffic accident.

The results achieved are very interesting and promising.

English language and style are fine.

The paper needs however some improvements to make it more suitable for publication.

Thank you very much for your positive feedback!

The reviewer would suggest these main actions:

Improve the structure of the paper. In its present form the paper is rather unbalanced. Introduction occupies a too little section: although the literature review presents a good level of details, it would be appropriate to add some papers, from major international transportistic journals, on the relationship between accident density, and its severity with a variety of independent variables practically ignored in this study.

We have searched for more references following your indications and added more information in the Introduction.

Highlight the limitations: the model is strongly conditioned by the specific country analysed in the study and by the absence of many other parameters that influence the gravity of a crash.

These statements have been included in Limitations in the revised version of the text.

Strengthen the conclusions: authors should be better explaining how this implemented model can be used to design interventions to reduce mortality from traffic accidents.

We have followed your suggestion in the Conclusion.

In the next studies, it would be important to consider new fundamental parameters such as: types of road (not only urban or interurban), types of road section (bridge, tunnel, embankment, trench, etc.), speed limits and speed collision.

Thank you. This has been included in Limitations and in Implications to research.

Round 2

Reviewer 1 Report

The authors have addressed the concerns raised in my report. I am satisfied that the explanation with Figure 1 is sufficient and no more needs to be added to the paper to clarify this point.

I have reviewed Figures S1 – S3 supplied in response to my request for a comparison of the linear and spline fits to the data. However two of these figures have not served the authors’ case well. The spline curve for Age shows a bump at age 85, which is surely an artefact of the scarcity of data at older ages. The spline curve for age of the car runs all the way out to 120 when surely there is no data at all beyond about 40. The authors should rescale this graph to be consistent with the data.

I am therefore recommending a minor revision to address these graphs before the article is accepted for publication.

Author Response

The authors have addressed the concerns raised in my report. I am satisfied that the explanation with Figure 1 is sufficient and no more needs to be added to the paper to clarify this point.

Thank you very much for your comments.

I have reviewed Figures S1 – S3 supplied in response to my request for a comparison of the linear and spline fits to the data. However two of these figures have not served the authors’ case well. The spline curve for Age shows a bump at age 85, which is surely an artefact of the scarcity of data at older ages. The spline curve for age of the car runs all the way out to 120 when surely there is no data at all beyond about 40. The authors should rescale this graph to be consistent with the data.

We have followed your suggestion with the graphs.

I am therefore recommending a minor revision to address these graphs before the article is accepted for publication.

Thank you very much for your valuable feedback!

Reviewer 3 Report

Thanks for taking into account my suggestions.

Author Response

Thanks for taking into account my suggestions.

Thank you very much for for your valuable feedback!
